# A Comparative Study on Recent Automatic Data Fusion Methods †

Luis Manuel Pereira, Addisson Salazar * and Luis Vergara

Instituto de Telecomunicaciones y Aplicaciones Multimedia, Universitat Politècnica de València,
46022 Valencia, Spain; lmpergon@doctor.upv.es (L.M.P.); lvergara@dcom.upv.es (L.V.)
* Correspondence: asalazar@dcom.upv.es
† This paper is an extended version of our paper published in 17th International Work-Conference on Artificial
Neural Networks (IWANN2023), Ponta Delgada, Portugal, 19–21 June 2023.

**Abstract:** Automatic data fusion is an important field of machine learning that has been increasingly studied. The objective is to improve the classification performance from several individual classifiers in terms of accuracy and stability of the results. This paper presents a comparative study on recent data fusion methods. The fusion step can be applied at early and/or late stages of the classification procedure. Early fusion consists of combining features from different sources or domains to form the observation vector before the training of the individual classifiers. On the contrary, late fusion consists of combining the results from the individual classifiers after the testing stage. Late fusion has two setups, combination of the posterior probabilities (scores), which is called soft fusion, and combination of the decisions, which is called hard fusion. A theoretical analysis of the conditions for applying the three kinds of fusion (early, late, and late hard) is introduced. Thus, we propose a comparative analysis with different schemes of fusion, including weaknesses and strengths of the state-of-the-art methods studied from the following perspectives: sensors, features, scores, and decisions.

**Keywords:** data fusion; early fusion; late fusion; late hard fusion; decision fusion

## 1. Introduction

Currently, information is growing exponentially in complexity, volume, variety, and veracity. We can extract and derive valuable information from data to learn about the nature of things [1–3]. The recent development of machine learning (ML) methods, in conjunction with the increasing capacity of sensor devices, has posed a critical problem to efficiently solve data classification. This problem comprises two questions: how to efficiently combine information from several sources or modalities of data, i.e., multimodal measurements, and how to combine the results from several classifiers. This latter question assumes that there are relationships between the random variables of the score distributions provided by each of the classifiers to be combined that allow the results to be improved.

In general, processes that require parameter estimation from multiple sources can benefit from data fusion in a context of multiple classifiers. Data fusion aims to combine the information received from the real world to make the results more heterogeneous and informative than the original ones [4]. In this way, the objectives are to increase the reliability of the classification and the quality of the extracted information. Data fusion aims to improve classification accuracy by combining the predictions of multiple models or classifiers to obtain a more robust final classification. Thus, the individual classifier biases and generalization for data not seen in classifier training can be solved. Data fusion aims to effectively handle noisy data, adapt to different changing scenarios and conditions, and improve the stability of problem solving [5,6].

The exponential expansion of data in recent years due to new technologies has also brought with it an increase in the interest of researchers in data fusion, driven by the need to extract relevant information from increasingly complex datasets [7–9]. Let us show a

descriptive example of the objectives of decision fusion. Figure 1 shows a toy example of the decision boundaries formed by different methods of classification: linear discriminant analysis (LDA); quadratic discriminant analysis (QDA); and a Bayesian classifier implementing non-parametric probability estimation. In addition, the boundaries obtained by the fusion of the three single classifiers are also shown. The problem outlined in Figure 1 is two-class classification, i.e., to distinguish between the samples of classes labeled 0 and 1. The geometry of the data is 3D, considering three features of the multidimensional vector observation.

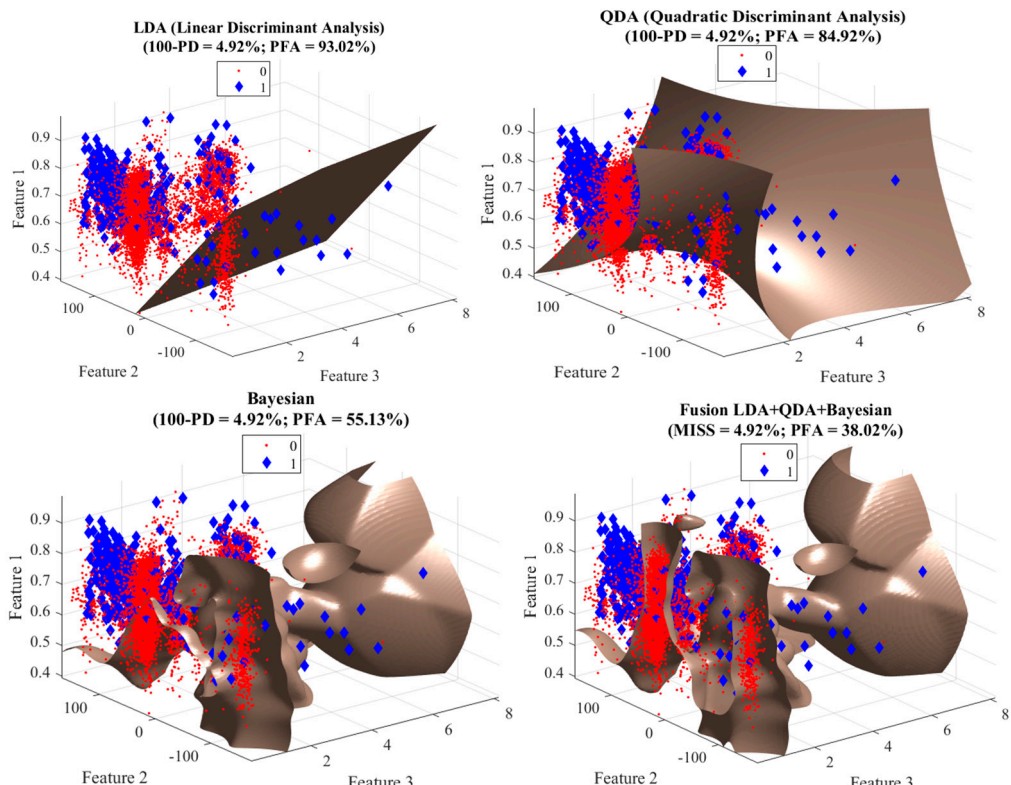

**Figure 1.** Decision boundaries established by different classifiers and the fusion of the classifiers.

We can see that the data of the two classes are squeezed together, and it is difficult to separate them. The separation boundaries of the three classifiers are different: that provided by LDA consists of a hyperplane; QDA's is a paraboloid; and the boundary provided by the Bayesian classifier describes a complex surface. The efficiency of those surfaces to separate the data of the two classes in terms of missed detections (100-probability of detection (PD)) and probability of false alarm (PFA) is shown. Setting the value of the miss detection at 4.92%, it is clearly shown that the best PFA is obtained by the Bayesian classifier (53.13%) consistently with the classifier boundary decision surface that best fits the data. However, the fusion of the three classifiers yields a better suitable separation surface, which is recorded in a much lower PFA (38.02%).

In order to perform the comparative study, we propose the following four schemes to analyze the state of the art of data fusion methods: (i) early fusion from sensors; (ii) early fusion from features; (iii) late fusion from scores; and (iv) late fusion from decisions. The comparison of the methods studied considers aspects such as complexity, optimization, accuracy, and volume of data. Moreover, a theoretical analysis of the conditions for applying the different kinds of fusion is presented.

*Data Fusion Concepts*

Data fusion techniques attempt to combine multiple sources of information to achieve accuracy and precision in decision-making that would not be possible to achieve with

the use of a single source of information in isolation. The fusion of multiple sources of information, in addition to adding a certain redundancy that provides reliability and robustness, can provide complementary information with which to increase performance and accuracy in the decision-making process. For simplicity, we will discuss the two-class classification (detection) problem. Figure 2 shows a diagram of the fusion process from a statistical standpoint. It shows how various sources of information (with different probability density functions (PDFs) for the two hypotheses—$H_1$: detection and $H_0$: non-detection) are merged into a single one. The separation between hypotheses is improved, i.e., there is less data confusion in the fusion result.

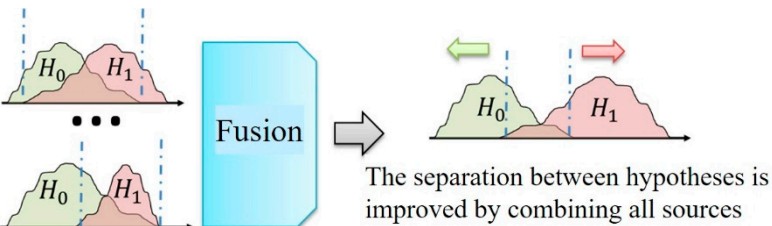

**Figure 2.** Improving detection capabilities by fusing information.

We can find areas or applications where it is common to work with various sources of information and, thus, a combination of information is necessary. For example, distributed sensing systems [10] use a multi-sensor system composed of a set of identical sensors placed with a certain spatial distribution. Multimodal systems [11], where sources of information of various kinds from different sensors and/or sources of unsensed information (such as web resources, databases, metadata, etc.) are used, are common in biometric systems or multimedia analysis problems. There are also detection techniques, such as expert combination [12], where an attempt is made to combine a series of simple detectors to achieve results that would be very difficult to achieve with a single detector and/or reduce the complexity and computational resources required to implement it.

The random variables that characterize the data from each information source can follow different probability distributions. It may be because they are data from different types of sensors that capture different phenomena produced by the same event, or from sensors of the same type, but whose position relative to the source that produces the event translates into different distributions of their observations. It is very common to use data that come from the extraction of multiple features related to various physical aspects or that are of a different nature from a single flow of information provided by a single sensor. In other cases, the aim is to combine different algorithms or processing techniques with very different output information.

The use of multiple sources and redundant information can provide complementary information that can be exploited by combining all available sources. Thus, the discriminatory information on the occurrence or not of the event to be detected can be increased, and both overall detection performance, robustness, and reliability can be improved. The sources of information generally share a common origin, in which the event to be detected originates, so it is common to find the existence of statistical dependence between them. Statistical dependency between sources can introduce complementary information that we can benefit from to improve detection performance.

Data fusion is a booming area of research. There are several quite different real-world problems where data fusion has been recently applied, including the following: detecting breast tumors in tomosynthesis images [13]; video summarization [14]; multi-scenario violence detection [15]; energy-efficient grid-based routing protocols [16]; image sparse representation [17]; accurate skin lesion classification [18]; multimodal inference of mental workload for cognitive human machine systems [19]; fingerprint and online signature for multimodal biometrics [20]; and image fusion using per-pixel saliency visualization [21].

Different levels of fusion can be defined depending on the stage of the detection process in which the integration of the information is carried out. A detection problem can generally be divided into four stages (Figure 3). The first of these is the sensing stage, where one or more sensors are responsible for obtaining a series of measurements of the environment where the event occurs. These raw data (*y*) are processed to extract certain characteristics (features) of the event (*x*), from which detectors or binary classification algorithms will be able to yield data sample scores (*z*), usually related to the probability that the event to be detected has occurred. After thresholding scores, a set of binary decisions (*u*) is obtained on the occurrence or not of the event.

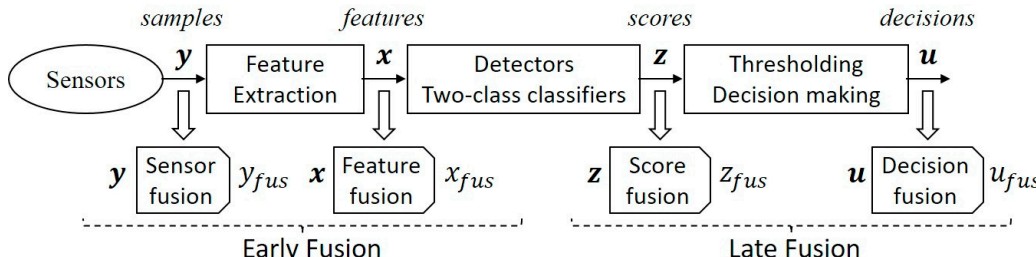

**Figure 3.** Outline of a data fusion pipeline in a detection system.

Thus, the sensor fusion level relates to when the different sample flows provided by sensors are combined ($y \rightarrow y_{fus}$); the feature fusion level is where the raw data features extracted are fused ($x \rightarrow x_{fus}$); the score fusion level concerns obtaining fused scores from the ones obtained by the classifiers ($z \rightarrow z_{fus}$). Those scores are thresholded to obtain decisions that are fused at the decision level ($u \rightarrow u_{fus}$). The levels of sensor and feature fusion can be grouped into what is known as early or pre-detection fusion. The levels of score and decision fusion are often referred to as late fusion or detector fusion [22–24]. The different levels of fusion are not mutually exclusive, so systems can be designed that combine fusion at different levels based on information provided by different sources throughout all stages.

Depending on the type of data to be combined, we can classify the fusion into two categories, soft fusion and hard fusion. (i) Soft fusion refers to the combination of continuous data, which can be modeled by continuous random variables, usually characterized by PDFs under each of the hypotheses. Within the detection process, we can generally find soft information in the data provided by the sensors (*y*), the features extracted from them (*x*), and in the scores provided by some types of detectors (*z*). (ii) Hard fusion combines discrete data, which are modeled by discrete random variables under each of the hypotheses. Hard information fusion is commonly associated only with the detector fusion stage, where binary random variables are used.

## 2. Early Fusion/Late Fusion Comparison

### 2.1. Early Fusion

Let us assume for simplicity the scenario of two classes ($k = 1, 2$), and two sets of features $x_1$, $x_2$, $x_1$, $x_2$, from two different modalities that we will assume conditionally independent, i.e., $p(x_1, x_2/k) = p(x_1/k)p(x_2/k)$. Suppose we perform early fusion, choosing the class that maximizes the a posteriori probability given the two sets of features $x_1$, $x_2$ i.e., the decision rule will be:

$$P(k = 1/\mathbf{x}_1, \mathbf{x}_2) \underset{k=2}{\overset{k=1}{\gtrless}} P(k = 2/\mathbf{x}_1, \mathbf{x}_2) \quad \Leftrightarrow \quad P(k = 1/\mathbf{x}_1, \mathbf{x}_2) \underset{k=2}{\overset{k=1}{\gtrless}} 0.5, \tag{1}$$

where we have taken into account that $P(k = 1/\mathbf{x}_1, \mathbf{x}_2) + P(k = 2/\mathbf{x}_1, \mathbf{x}_2) = 1$. Note that rule (1) implies minimization of the probability of error if we assume that the costs of being wrong in the decision are symmetric and normalized to 1. That is, considering that we had

exact knowledge of $P(k = 1/\mathbf{x}_1, \mathbf{x}_2)$, no other fusion rule allows us to reduce the probability of error more than (1).

### 2.2. Late Soft Fusion

Suppose now that we perform late soft fusion. For this purpose, we generate an a posteriori probability (score) separately for each modality: $s_1 = P(k = 1/\mathbf{x}_1)$ and $s_2 = P(k = 1/\mathbf{x}_2)$. Note that when it is the case that $P(k = 1/\mathbf{x}_1) + P(k = 2/\mathbf{x}_1) = 1$ and $P(k = 1/\mathbf{x}_2) + P(k = 2/\mathbf{x}_2) = 1$, it is enough to consider the scores $s_1$ and $s_2$ defined. We then generate a new score $s$ fusing $s_1$ and $s_2$ through a certain fusion function $s = f(s_1, s_2)$ $\quad 0 \leq s \leq 1$ and the new decision rule will be:

$$s \underset{\underset{k=2}{<}}{\overset{\overset{k=1}{>}}{}} 0.5, \tag{2}$$

Note that $s = f(s_1, s_2) = f(P(k = 1/\mathbf{x}_1), P(k = 1/\mathbf{x}_2))$ is ultimately a function of multivariate random variables $\mathbf{x}_1$, $\mathbf{x}_2$, which will generally be different from $P(k = 1/\mathbf{x}_1, \mathbf{x}_2)$ in (1) and therefore rule (2) can never achieve a probability of error lower than rule (1), assuming knowledge of $P(k = 1/\mathbf{x}_1, \mathbf{x}_2)$. For instance, let us analyze the case of the fusion method based on $\alpha$-integration [25–28]. First, we apply Bayes' rule:

$$s_1 = P(k = 1/\mathbf{x}_1) = \frac{p(\mathbf{x}_1/k=1)P_1}{p(\mathbf{x}_1)}; \quad s_2 = P(k = 1/\mathbf{x}_2) = \frac{p(\mathbf{x}_2/k=1)P_1}{p(\mathbf{x}_2)}, \tag{3}$$

We will now consider the $\alpha$-integration of both scores to obtain $s$:

$$
\begin{aligned}
s &= \left( w_1 s_1^{\frac{1-\alpha}{2}} + w_2 s_2^{\frac{1-\alpha}{2}} \right)^{\frac{2}{1-\alpha}} = \left( w_1 \left( \frac{p(\mathbf{x}_1/k=1)P_1}{p(\mathbf{x}_1)} \right)^{\frac{1-\alpha}{2}} + w_2 \left( \frac{p(\mathbf{x}_2/k=1)P_1}{p(\mathbf{x}_2)} \right)^{\frac{1-\alpha}{2}} \right)^{\frac{2}{1-\alpha}} = \\
&= \left( \frac{w_1 (p(\mathbf{x}_1/k=1)p(\mathbf{x}_2)P_1)^{\frac{1-\alpha}{2}} + w_2 (p(\mathbf{x}_2/k=1)p(\mathbf{x}_1)P_1)^{\frac{1-\alpha}{2}}}{(p(\mathbf{x}_1)p(\mathbf{x}_2))^{\frac{1-\alpha}{2}}} \right)^{\frac{2}{1-\alpha}} = \\
&= \frac{\left( w_1 \left( p(\mathbf{x}_1/k=1)p(\mathbf{x}_2)w_1^{\frac{2}{1-\alpha}} \right)^{\frac{1-\alpha}{2}} + w_2 \left( p(\mathbf{x}_2/k=1)p(\mathbf{x}_1)w_2^{\frac{2}{1-\alpha}} \right)^{\frac{1-\alpha}{2}} \right)^{\frac{2}{1-\alpha}}}{p(\mathbf{x}_1)p(\mathbf{x}_2)} P_1
\end{aligned} \tag{4}
$$

On the other hand, the score generated in rule (1) can be written as:

$$
\begin{aligned}
P(k = 1/\mathbf{x}_1, \mathbf{x}_2) &= \frac{p(\mathbf{x}_1,\mathbf{x}_2/k=1)P_1}{p(\mathbf{x}_1,\mathbf{x}_2)} = \frac{p(\mathbf{x}_1/k=1)p(\mathbf{x}_2/k=1)}{p(\mathbf{x}_1,\mathbf{x}_2)} P_1 = \\
&= \frac{p(\mathbf{x}_1/k=1)p(\mathbf{x}_2/k=1)}{p(\mathbf{x}_1)p(\mathbf{x}_2/\mathbf{x}_1)} P_1
\end{aligned} \tag{5}
$$

where we have taken into account the assumed conditional independence between the characteristics of the two different modalities $\mathbf{x}_1$, $\mathbf{x}_2$. Comparing (4) and (5), we can conclude an interpretation of the $\alpha$-integration as an attempt to approximate the optimal score in (5). On the one hand, in relation to the denominator, when operating the two modalities through separate channels before merging, it is not taken into account that $\mathbf{x}_1$ and $\mathbf{x}_2$ are (unconditionally) dependent. On the other hand, we can see that the parameters $\alpha$, $w_1$ and $w_2$, which are estimated by minimizing a certain cost function (e.g., error probability), should tend to be adjusted so that $p(\mathbf{x}_2)w_1^{\frac{2}{1-\alpha}} \simeq p(\mathbf{x}_2/k = 1)$ and $p(\mathbf{x}_1)w_2^{\frac{2}{1-\alpha}} \simeq p(\mathbf{x}_1/k = 1)$, in order to obtain the minimum probability of error of rule (1). In any case, it is clear that, assuming perfect knowledge of late soft fusion can never provide a lower probability of error than early fusion. However, we must also consider that the use of particular classifiers is conditioned to the requirements of the size of the training datasets, and thus conditioning the fusion of the classifiers. The estimation of the sample size for training to obtain an estimated error is a complex task; see for instance the proxy learning curve for the Bayes classifier [29].

### 2.3. Late Hard Fusion

In this case, first, the binary decisions of each modality are generated, and from them a decision fusion rule is established to generate the final decision. Let us denominate $d_1$, $d_2$ the binary variables corresponding, respectively, to the binary decisions of each modality, i.e., we can write:

$$d_1 = u(s_1 - 0.5); \quad d_2 = u(s_2 - 0.5), \tag{6}$$

where $u(x) = \begin{cases} 1 & x > 0 \\ 0 & x < 0 \end{cases}$ is the step function. As we can see, the binary variables are equal to 1 if we choose class 1 and equal to 0 if we choose class 2. The final decision will be the result of applying a fusion rule to $d_1$, $d_2$. In the case of two modalities, we have only two possible reasonable rules:

$$d = g_1(d_1, d_2) = \begin{cases} 1 & \text{si } d_1 = 1, \ d_2 = 1 \\ 0 & \text{all other cases} \end{cases} \qquad d = g_2(d_1, d_2) = \begin{cases} 0 & \text{si } d_1 = 0, \ d_2 = 0 \\ 1 & \text{all other cases} \end{cases}, \tag{7}$$

It seems natural that late hard fusion, in exchange for its simplicity, leads to worse performance than late soft fusion, since binarization prior to fusion entails the loss of information. Somewhat heuristically, we can reason as follows. Suppose that the late soft fusion function $f(s_1, s_2)$ is optimized to minimize a certain cost function of the binary classifier, such as the probability of error. If we take into account that $d_1$, $d_2$ are a function of the scores $s_1$, $s_2$, we can think that ultimately the functions $g_1(d_1, d_2)$ and $g_2(d_1, d_2)$ in (7) are functions of $s_1$, $s_2$, which will, in general, be different from the optimal function used in late soft fusion and which will, therefore, not minimize the chosen cost function. To be more specific, let us consider again the integration. It is easy to check that:

$$\alpha \to \infty \quad \Rightarrow \quad s \to \min(s_1, s_2) \quad \alpha \to -\infty \quad \Rightarrow \quad s \to \max(s_1, s_2), \tag{8}$$

But the following fusion rules are equivalents:

$$\min(s_1, s_2) \underset{\substack{< \\ k=2}}{\overset{\substack{k=1 \\ >}}{\gtrless}} 0.5 \quad \Leftrightarrow \quad g_1(d_1, d_2) \qquad \max(s_1, s_2) \underset{\substack{< \\ k=2}}{\overset{\substack{k=1 \\ >}}{\gtrless}} 0.5 \quad \Leftrightarrow \quad g_2(d_1, d_2), \tag{9}$$

That is, we can interpret late hard fusion as a late soft fusion for a particular choice of integration parameters of $\alpha$-integration, which will not, in general, be those that minimize the cost function and will therefore lead to worse (or, at best, equal) performances than those achievable with late soft fusion.

In short, we conclude that, if the a posteriori probability or score $P(k = 1/\mathbf{x}_1, \mathbf{x}_2)$ is known precisely, late fusion will not be able to outperform early fusion. On the other hand, the hard fusion will not be able to outperform late soft fusion either, as we have justified above.

The exact knowledge of $P(k = 1/\mathbf{x}_1, \mathbf{x}_2)$ is only strictly possible if we have an infinite training set and we assume statistical consistency, i.e., $\hat{P}(k = 1/\mathbf{x}_1, \mathbf{x}_2) \overset{N \to \infty}{\to} P(k = 1/\mathbf{x}_1, \mathbf{x}_2)$, where $N$ is the size of the training set of each class (for simplicity we assume the same for both classes). From a practical point of view, accurate estimation implies having sufficiently large training sizes for each class. Unfortunately, it is not easy to determine in general what the adequate minimum size is. This is outside the scope of this paper. However, there are recent works that provide certain criteria in this regard [30].

### 3. Early Fusion from Sensor and Features

In Sections 3 and 4 we will study the state-of-the-art data fusion methods and briefly describe them comparatively according to different points in the data processing where fusion can take place. The study includes the analysis of strengths and weaknesses of the methods and related works. Figure 4 shows the proposed outline to explain the different fusion approaches.

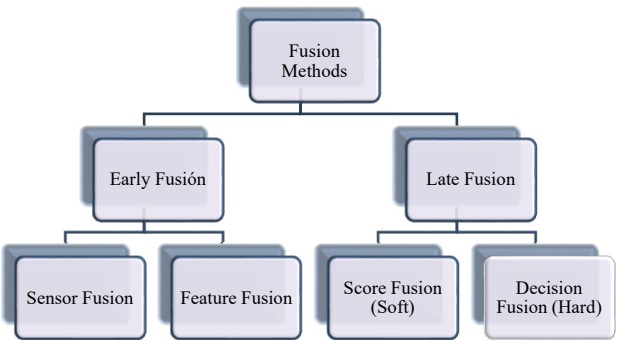

**Figure 4.** Diagram of fusion levels.

### 3.1. Early Fusion from Sensor

The process of combining disparate sensor data is known as sensor fusion. Figure 5 shows a schematic illustrating early sensor fusion. There are several ways of acquiring data from the sensors:

✓ Individual sensor, multiple samples.

This first case is the simplest as it involves a single sensor from which we obtain multiple samples.

✓ Multisensors.

The essence of this data acquisition is to combine information from multiple sensors into a single sensor. In this way, greater accuracy can be achieved than could be achieved using a single sensor [31].

✓ Multimodal.

Multimodal acquisition integrates information from multiple sources to compensate for each of them. An example is the simultaneous acquisition of electroencephalogram (EEG) and functional magnetic resonance imaging (fMRI) data. On the one hand, we have a high temporal resolution, as in the case of EEG, and on the other hand, we have fMRI data with a high spatial resolution [32].

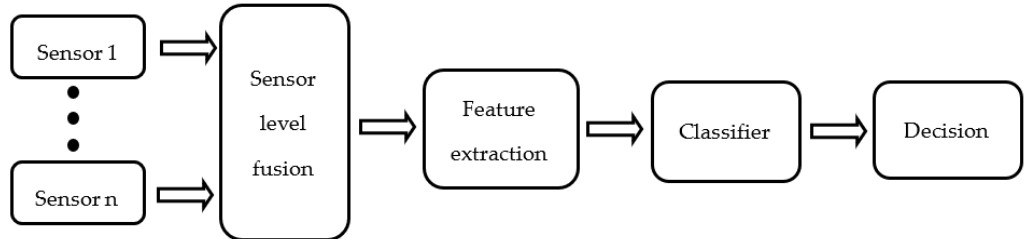

**Figure 5.** Early sensor-level fusion scheme.

At the sensor level, there are several methods of data fusion. Among them, we can find the Kalman Filter, Bayesian Inference, Fuzzy Logic, Artificial Neural Networks (ANNs) and Dempster–Shafer (DS). In Table 1, we present a brief description of these methods, and finally, we offer a comparative table where we will find the weaknesses and strengths of each of these methods and related works.

✓ Kalman Filter.

The Kalman filter is an ideal recursive statistical data processing algorithm that continuously computes an estimate of a continuous-valued state from periodic observations [33]. It uses an explicit statistical model of how the vector of interest $x(t)$ changes over time and an explicit statistical model of how the vector of observations $z(t)$ made are related [1]. The gains used in a Kalman filter are chosen to ensure that the resulting estimate minimizes the

mean square error with certain assumptions about the observation models and processes used [34].

✓ Bayesian Inference.

Bayesian inference is a statistical data fusion algorithm based on Bayes' theorem of a posteriori probability for estimating an n-dimensional state vector $Z$, given the observation or measurement $X$ [35].

According to Bayes' theorem, given a set of measurements $X_k = \{x_1; \ldots; x_k\}$ in time $k$, the Bayesian interference is formulated as:

$$p\left(z_k \middle| X^k\right) = \frac{p(x_k \vee z_k) p\left(z_k \middle| X^{k-1}\right)}{p\left(X^k \middle| X^{k-1}\right)}, \tag{10}$$

where $p(x_k \vee z_k)$ is the probability function based on a given sensor measurement model, $p\left(z_k \middle| X^{k-1}\right)$ is the a priori probability distribution, and $p\left(X^k \middle| X^{k-1}\right)$ is the probability density function considering all hypotheses.

If we assume that the statistical independence between the set of measurements $X_k = \{x_1; \ldots; x_k\}$, then we can combine them to infer the state of the observed system [36]. This method requires knowledge of the a priori probability distribution of the states.

✓ Fuzzy Logic.

Fuzzy Logic is a logic that uses neither wholly true nor false expressions. It is applied to concepts that can acquire any value of veracity within a set of values that oscillate between two extremes: absolute truth and total falsehood [37].

The main benefits of using fuzzy logic techniques are the simplicity of the approach, its ability to handle ambiguous information, and its capacity to include heuristic knowledge about the phenomenon under consideration [2,38]. The use of fuzzy logic for sensor fusion has demonstrated a high degree of accuracy and precision, but the complicated computations required are an obstacle [39].

✓ Artificial Neural Networks (ANNs).

ANNs are mathematical models of non-linear computational elements operating in parallel and linked together in a topology distinguished by several ponderable connections. Comparing ANN analysis with conventional linear or non-linear analysis, it has proven to be a more powerful and flexible method [40].

One of the features of ANNs is their adaptive learning capability. Developing a priori models or specifying probability distribution functions is unnecessary, because neural networks can learn differentially through examples and training [41].

A typical ANN structure comprises three main layers: the input layer, one or more hidden caps, and the output layer. One or more neuron-like nodes distinguish each. The number of input and output variables determines the number of neurons in the input and output layers. Depending on the problem's difficulty, the number of hidden layers and the number of neurons associated with each hidden layer varies [42].

✓ Dempster–Shafer (DS).

Dempster–Shafer theory offers an alternative to the traditional probabilistic approach for the mathematical representation of uncertainty. It has been widely applied in various applications, such as target tracking, surveillance, robotic navigation, and signal and image processing [43].

As is well known, DS represents the uncertainty or imprecision in a hypothesis that characterizes all possible system states. A probability mass assignment (PMA) is applied to such a hypothesis, which results in a decision when combined. Therefore, creating a function for the mass assignment and combining it is essential for accurate prediction. By applying a combinatorial rule to the sources of evidence, DS achieves the goal of data fusion [9].

Table 1. Comparison between early sensor-level fusion methods and related works.

| Sensor-Level Fusion Methods | Strengths | Weaknesses | Works |
|---|---|---|---|
| Kalman Filter | The Kalman filter can provide highly efficient and accurate results in contexts where the system conditions are well understood and the models are correct. Estimates are performed recursively. This property makes it computationally efficient and suitable for real-time applications. | It is not a method designed for optimization; therefore, it cannot converge to local or global minima. It is intended for state estimation and prediction in dynamic systems. It requires a broad knowledge of probabilities, especially the subject of Gaussian conditionality in random variables. | [44–48] |
| Bayesian Inference | It is a recursive technique and can compute probabilities and posterior probabilities for multiple hypotheses. If the conditions are well understood and the models are correct, it offers a convenient setup for various models, such as hierarchical models and missing data problems. | The probability distribution of the states must be known a priori. It often comes with a high computational cost, especially in models with many parameters. As the size of the data increases, handling the distributions becomes more difficult. | [35,49–52] |
| Fuzzy Logic | Accurate results in non-linear and challenging-to-model processes. It is based on logical sets and reasoning that are easy to understand and, therefore, to use. Provides a simple mechanism for reasoning with vague, ambiguous, or imprecise information. | Extensive validation and verification of fuzzy algorithms are necessary. Accurately defining fuzzy sets or membership functions requires time and effort. In addition, increasing the dimension of the data makes it more challenging to model the problem. Fuzzy control systems depend on human experience and knowledge. | [2,39,53–55] |
| Artificial Neural Networks | It is self-learning and can execute tasks that a linear program cannot and is able to process unorganized data. Its structure is adaptive in nature. When an element of the neural network slows down, it can continue without problems, thanks to its parallel characteristics and is efficient at handling data noise, separating only the necessary information. | Requires prior training to operate, a large amount of data to achieve adequate efficiency and a lot of processing time for large neural networks. Requires specific hardware equipment to operate due to their computational complexity. If not handled properly, neural networks may be overfitted to the training data and not generalize well to new data. It can converge to local minima instead of global minima, although there are solutions to this problem, such as weight initialization and regularization in terms of L1 and L2. | [40,56–60] |
| Dempster–Shafer | Such a theory can provide accurate results if the evidence is accurate and reliable. Although there is no hard and fast rule regarding data, a limited amount of data can help manage uncertainty and consistently combine information. | Generally, it presents a high computational complexity, although this may vary depending on the amount of data, hypotheses, and uncertainty of the problem to be treated. | [9,61,62] |

### 3.2. Early Fusion from Features

These methods or techniques aim to reduce the feature vector's dimension while preserving as much information as possible. Figure 6 shows a schematic of the feature-level fusion methods.

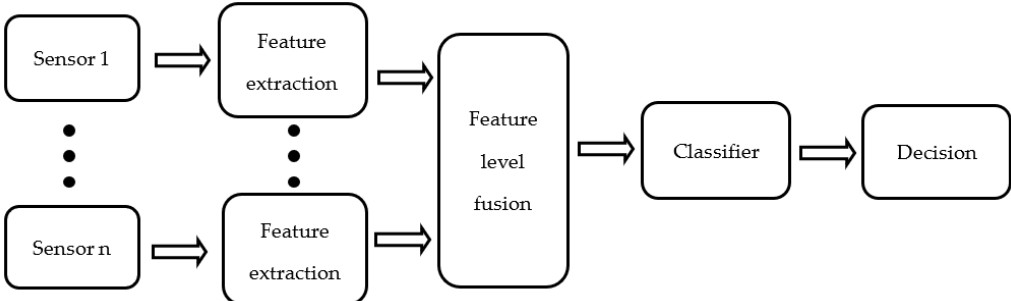

**Figure 6.** Early feature-level fusion scheme.

Among the feature-level fusion methods, we can find Principal Component Analysis (PCA), Singular Value Decomposition (SVD), Multidimensional Scaling (MDS), and Deep Learning. Below, in Table 2, we will briefly describe them, detailing their strengths and weaknesses and the related works that have been most cited.

✓ Principal Component Analysis (PCA).

PCA is a mathematical procedure transforming several correlated variables into un-correlated variables called principal components [63]. This technique reduces the data set (in this case, the features extracted from multiple sensors) while maintaining the statistical information and at the same time minimizing the loss of information [6]. Principal components are calculated as linear combinations of the original data or variables [64].

We refer to each main component as $Y_j$ and a set of variables $(X_1, X_2, \ldots, X_p)$. The first principal component is the normalized linear combination of these variables with the highest variance.

$$Y_1 = \phi_{11}X_1 + \phi_{21}X_2 + \cdots + \phi_{p1}X_p, \tag{11}$$

The normalized linear combination implies that:

$$\sum_{j=1}^{p} \phi_{j1}^2 = 1, \tag{12}$$

The terms $\phi_{11}, \ldots, \phi_{1p}$ define the component and are called loading. They can be interpreted as each variable's weight in each component.

✓ Singular Value Decomposition (SVD).

SVD is a powerful mathematical technique to decompose a matrix into constituent parts. It is a generalization of the autovalues decomposition of a matrix and can be applied to any rectangular matrix, not just square matrices. It helps to reduce data sets containing a large number of values and is useful for generalizing meaningful solutions with fewer values [65–67].

A matrix $X$ of $m \times n$ with m and n dimensions of the row and column vectors can be related to the diagonal matrix $\sum$ de $m \times n$, which satisfies the following equation:

$$\sum = U^T X V, \tag{13}$$

where $U$ is an orthogonal matrix of $m \times n$ and $V$ is an orthogonal matrix of $n \times n$. When rewritten, it is satisfied that:

$$X = U \sum V^T, \tag{14}$$

✓ Multidimensional Scaling (MDS).

MDS is a dimensionality reduction technique that converts multidimensional data into a lower dimensional space while maintaining intrinsic information [68]. This set of statistical techniques takes as input similarity estimates between a set of data [69]. This method is a much more flexible alternative to other multivariate analysis methods, since it only requires a matrix containing the similarity or dissimilarity between the input data.

In general, MDS takes a proximity matrix as input $\in M_{nxn}$, where $n$ is the number of stimuli and each element $\delta_{ij}$ of $\Delta$ represents the proximity between stimulus $i$ and stimulus j.

$$\Delta = \begin{bmatrix} \delta_{11} & \cdots & \delta_{1n} \\ \vdots & \ddots & \vdots \\ \delta_{n1} & \cdots & \delta_{nn} \end{bmatrix}, \tag{15}$$

From this proximity matrix, MDS provides us with an output matrix $\in M_{nxm}$, where n is the number of stimuli as before and m is the number of dimensions. Each value of $x_{ij}$ represents the coordinate of stimulus i in dimension j.

$$X = \begin{bmatrix} x_{11} & \cdots & x_{1m} \\ \vdots & \ddots & \vdots \\ x_{n1} & \cdots & x_{nm} \end{bmatrix}, \tag{16}$$

From the matrix X we can calculate the distance between any two stimuli by applying the general Minkowski formula:

$$d_{ij} = \left[ \sum_{t=1}^{m} \left( x_{it} - x_{jt} \right)^p \right]^p, \tag{17}$$

where p can be a value between 1 and infinity. From these distances, we can obtain a matrix of distances $D \in M_{nxn}$.

$$D = \begin{bmatrix} d_{11} & \cdots & d_{1n} \\ \vdots & \ddots & \vdots \\ d_{n1} & \cdots & d_{nn} \end{bmatrix}, \tag{18}$$

The solution provided by the MDS should be such that there is a maximum correspondence between the initial proximity matrix $\Delta$ and the matrix of distances obtained $D$.

✓ Deep Learning.

We can say that deep learning is a type of machine learning that aims to make an artificial intelligence (AI) capable of learning by itself and performing tasks similar to those performed by humans, such as image recognition, speech recognition, and making predictions, over time and with large amounts of data and processing with algorithms.

Deep learning algorithms are applied to ANNs structured into input, hidden, and output layers. Nowadays, feature fusion using deep learning is very fashionable, especially in the medical field [70,71] and in social networks [72,73].

**Table 2.** Comparison between early feature-level fusion methods and related works.

| Feature-Level Fusion Methods | Strengths | Weaknesses | Works |
|---|---|---|---|
| Principal Component Analysis | Reduces the complexity of the data and identifies the most important features. This method captures the directions of maximum variability in the data. This means that the most informative features are retained as valuable in data fusion where relevant information is sought to be preserved. After the transformation, the variance of the data is preserved. | It is necessary to choose the correct number of principal components needed for the data set to avoid some loss of information. Although the method works quickly for large data sets, it requires high computational complexity and memory requirements. | [6,64,74–76] |

**Table 2.** *Cont.*

| Feature-Level Fusion Methods | Strengths | Weaknesses | Works |
|---|---|---|---|
| Singular Value Decomposition | This mathematical technique is precious for reducing the dimensionality of the data, capturing the most relevant and distinctive feature information, and eliminating redundancies and noise. | SVD can be computationally expensive for large data sets, mainly when applied to high-dimensional arrays.<br>It only makes use of a single data set, and by default, the resulting dimension reduction cannot incorporate any additional information that may be relevant.<br>The accuracy of the data may decrease if the data patterns are not linear. | [65–67,77–79] |
| Multidimensional Scaling | The solutions are relatively accurate.<br>It can be used to fuse different types of data into a shared space, which is useful when the features of the sensors are different.<br>The method provides a visual representation of the data in a two- or three-dimensional space, which can help to understand patterns and relationships. | It does not allow quantifying the level of quality of the result. Since it is based on the relationship between dimensions or factors, evaluating this relationship in numbers is tough.<br>Can be computationally expensive for large data sets and may require iterative optimization.<br>As the data are projected into a lower dimensional space, there may be a loss of information, which could affect the quality of the fused data. | [68,69,80–83] |
| Deep Learning | It assists in trend and pattern detection and does not need human assistance, i.e., it makes its own decisions.<br>It can handle many multidimensional data and constantly improves the algorithm to achieve more accurate results. It can fuse data from multiple sources, such as images, text, and signals, into a single architecture, leveraging information from different modalities. | Requires a large amount of data for training, which is time-consuming and computationally complex. Therefore, more powerful computers are needed for it to work.<br>Limited data availability can affect performance, as large amounts of data are needed for effective training.<br>Can be trapped in local minima instead of reaching the best possible solution (global minimum). | [70–73,84] |

## 4. Late Fusion from Scores and Decisions

### 4.1. Late Fusion from Scores (Late Soft Fusion)

Information from different detectors or classifiers is often referred to as scoring [85]. Score-level fusion is usually preferred because it offers the best compensation in terms of information content and ease of fusion [86,87]. Figure 7 shows a schematic of the workflow for these methods.

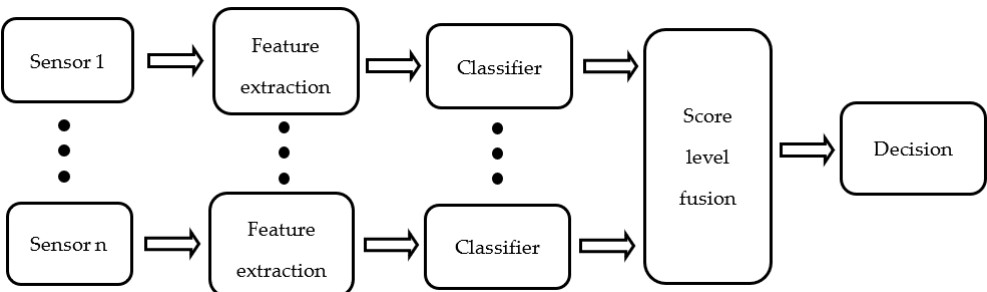

**Figure 7.** Late score-level fusion scheme.

According to the literature, Sum Rules, Likelihood Ratio (LR), Fusion by classifiers, Alpha Integration, and Behavior-Knowledge Space (BKS) are widely used score fusion methods. These techniques are described below, and Table 3 with the strengths and weaknesses of these methods and related work is presented.

✓ Sum and Weighted Sum Rules.

The Sum Rule is a simple fusion technique that operates directly on the raw data of the match scores [88]. It is given by the following equation:

$$fs = s_1 + s_2 + \cdots s_n, \tag{19}$$

where $fs$ is the fused score and $s_i$ represent the scores results. The weighted sum is similar, except that a weighting is assigned $w_i$ to each score depending on performance. It can be calculated as:

$$fs = w_1 s_1 + w_1 s_2 + \cdots w_n s_n, \tag{20}$$

✓ Likelihood Ratio (LR).

LR is a fusion approach based on the Neyman–Pearson theorem. This method does not require parametric adjustments and achieves a maximum valid acceptance rate (GAR) [89]. This approach is density-based and can achieve optimal performance at any desired false acceptance rate (FAR) point, provided that score density is accurately calculated.

The LR method reduces the feature values into a single metric (score) representing the similarity between the comparison items [90]. The genuine and impostor match score vectors are modeled using Gaussian mixture models. The likelihood ratio is calculated as the ratio of the genuine probability density over the impostor probability density.

✓ Fusion by classifiers.

Score fusion using classifiers is a technique in which classification models are used to combine scores. It aims to improve the accuracy and reliability of final decisions by taking advantage of the diversity of information provided by different classifiers.

The choice of classifier depends on the nature of the input scores and the application in question. It can be a simple classifier, such as logistic regression, or a more complex one, such as random forest (RF), support vector machine (SVM), and quadratic discriminant analysis (QDA). In addition, the right fit between performance and generalization must be achieved.

✓ Alpha Integration.

Alpha integration is a group of integrators containing many combinations, as may be the particular case of the alpha parameter. This approach was first proposed by Amari [91] to integrate multiple stochastic models by minimizing their alpha divergence. It has also been used to perform optimal integration of scores in binary classification problems [10].

This method was extended to integrate multiclass classifiers by considering the scores of each class in isolation in a technique called separated score integration (SSI) [26]. A new alpha integration method for late fusion of multiple classifiers that considers the combined effect of all classes in the multiclass problem called vector score integration (VSI) is proposed in [27].

✓ Behavior-Knowledge Space (BKS).

The BKS method is a trainable combination scheme at the abstract level, requiring neither measurements nor ordered sets of candidate classes. It attempts to estimate a posteriori probability by calculating each class's frequency for each possible set of classifier decisions based on a given training set [92].

In BKS, each classifier can assign a sample to one of the M possible classes. Each unit of a BKS represents a particular intersection of the decisions of an individual classifier. Thus, all possible combinations of the decisions of the individual classifiers are represented [8].

**Table 3.** Comparison between late score-level fusion methods and related works.

| Score-Level Fusion Methods | Strengths | Weaknesses | Works |
|---|---|---|---|
| Sum Rule | It does not require training samples. No sample distribution modeling is required. The addition process is fast and computationally efficient. This method works well for significant data inputs. | Requires estimation of normalized parameter and weights vector, and its accuracy is rarely consistent. It requires that match scores be of the same nature. It assumes comparable scales and strengths for input match scores | [88,93–96] |

**Table 3.** *Cont.*

| Score-Level Fusion Methods | Strengths | Weaknesses | Works |
|---|---|---|---|
| Likelihood Ratio | It has the potential to converge to maxima when maximizing the likelihood. It is able to handle discrete values in the score distribution. It does not involve the normalization of the score vector but the transformation of its respective likelihood ratio. If the densities of the scores are accurate, an optimum level is reached at any desired value of false acceptance rate (FAR). | Requires detailed modeling of score distributions. It is complex to implement due to the estimation of densities and is computationally complicated. It is very time-consuming as it involves a large amount of training samples. Requires a high knowledge of statistical techniques. | [89,90,97–100] |
| Fusion by classifier | It increases the overall accuracy of predictions by combining the strengths of different algorithms and reducing their weaknesses. Reduces the bias inherent in any algorithm and achieves greater flexibility by adapting to different data patterns. | Data fusion usually requires more data to train and validate the classifiers. It can increase computational complexity, especially if the data sets are large. Convergence to local and global minima is related to the type of classifier to be used. | [101–105] |
| Alpha Integration | It integrates many classic fusion operators and classifiers, optimizing fusion parameters and achieving more results that are accurate. | Optimizing the parameters is done by the gradient method, which may not converge to the global optimum. This method would inherit the weaknesses of the optimization method used. | [10,26,27,91,106] |
| Behavior-Knowledge Space | It does not depend on a prior hypothesis, such as statistical independence between classifier outputs. It allows the creation of a knowledge model that can organize and represent the knowledge extracted from the classifiers' prior behavior. | A limitation of this model is that with increasing data size the memory requirements increase exponentially. | [107–110] |

### 4.2. Late Fusion from Decisions (Late Hard Fusion)

Decision-level fusion aims to combine the decisions made by different classifiers to reach a common consensus and obtain a more accurate decision. Figure 8 shows where late fusion is performed at the decision level.

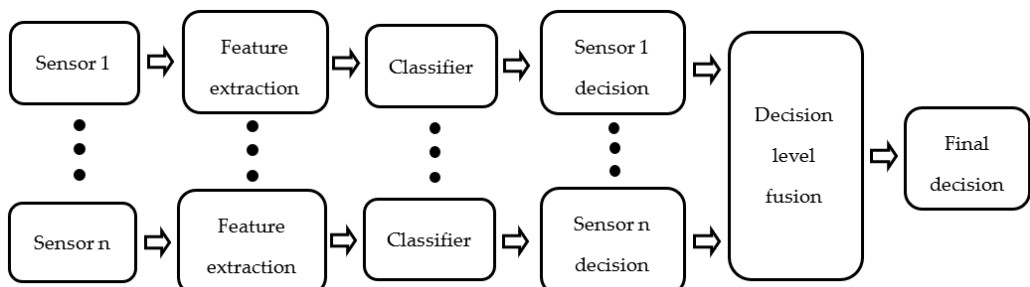

**Figure 8.** Late decision-level fusion scheme.

Majority Voting, Bagging, Boosting and Copula fusion are the most commonly used decision-level fusion methods. Table 4 below presents a description of these techniques, as well as the work related to these methods and a comparison taking into account their strengths and weaknesses.

✓ Majority Voting.

This type of method is based on a weighted process that combines the decisions provided by the classifiers. The simplest and most intuitive approach is based on voting by the predominant class, i.e., assigning a sample based on the most frequent class. In case of a tie, the sample is not classified. Protective approaches derive from applying specific limits, meaning that a sample is assigned only if the frequency of assignments to a class is greater than a selected threshold [111,112].

✓ Bagging.

It was originally introduced in 1996 [113] and expanded by [114]. It is based on sampling in its simplest form. The idea here is to sample subsets of the training set, generating independent bootstrap random replicates. The classifier is then constructed on each of these bootstrap samples, and finally, the constituent classifiers are aggregated by a simple majority vote [115].

The strength of bagging lies in unstable classifiers. These can be sensitive to minor alterations in the data set. Training the same classifier with slightly different data can give substantially different classifiers. Bootstrap sampling provides small random perturbations of the data set [116].

✓ Boosting.

Boosting has been proposed and perfected in the works of [117] which led to the most successful implementation called AdaBoost. Unlike the bagging method, which is based on random changes of the bootstrap sampling, the boosting method relies on classifiers built on weighted versions of the training set, which depend on previous classification results [115].

Generally, when there is little noise, boosting offers a reasonably accurate classifier; however, for situations with substantial noise, it is better to use other techniques [118,119] as previously shown.

✓ Copula fusion.

The theory of copulas relates multivariate distribution functions to their univariate marginal distribution functions and is explained by Sklar's theorem [120].

Considering an m-dimensional distribution function with marginal distribution functions $F_1, \ldots, F_m$, a copula function $C$ exists for all continuous random variables $x_1, \ldots, x_m$.

$$F(x_1, \ldots, x_m) = C(F_1(x_1), \ldots, F_m(x_m)) \tag{21}$$

The function $C$ is nothing more than a joint probability distribution of uniformly distributed random variables in the interval [0, 1].

**Table 4.** Comparison between late decision-level fusion methods and related works.

| Decision-Level Fusion Methods | Strengths | Weaknesses | Works |
|---|---|---|---|
| Majority Voting | Since this method is based on the linear combination of multiple detection algorithms, errors or misclassifications of one model do not affect the result. The excellent performance of the others can compensate for the poor performance of one classifier. It allows the results to be more robust and prone to overfitting. | It does not take into account the accuracy of the individual predictions of each classifier. If one classifier is more accurate, it will not have more influence than a less accurate one. Therefore, the result may be erratic. It should also be noted that the computational complexity could be high. | [111,121–123] |
| Bagging | Reduces variance and, in many cases, improves the accuracy of some predictors, especially if individual classifiers are prone to bias. Increases stability and eliminates the problem of overfitting for large amounts of data. | Introduces a loss of model interpretability; may experience biases when proper procedure is ignored. This method involves training and maintaining several models, which can significantly increase computational requirements compared to a single model. | [113–115,119] |

**Table 4.** *Cont.*

| Decision-Level Fusion Methods | Strengths | Weaknesses | Works |
|---|---|---|---|
| Boosting | Reduces variance and bias. Can generate a combined model that minimizes errors by avoiding the drawbacks of individual models. Weights those classifiers with better performance on the training data. Therefore, the accuracy of the model generally tends to improve. | It does not help solve the overfitting problem; on the contrary, it may increase it for large data sets. The computational complexity of the Boosting method can be considerable, especially in terms of training time and storage space. The number of iterations and the complexity of the base classifier are vital factors affecting complexity. | [115,117–119,124] |
| Copula fusion | It helps improve the fusion model's accuracy and generalization, especially when there are uncertain classifier outputs. If the choice and settings of copulas are correct, it converges to parameters that adequately represent the dependence between classifier outputs. | Depending on the copulas' complexity, the method may require complex computational calculations. Having sufficiently large data sets to train and evaluate the machine learning models is advisable. | [125–127] |

## 5. Conclusions

A comparative study on recent automatic data fusion methods has been presented. The fusion approaches were divided in four categories for ease of comparison: early fusion from sensors; early fusion from features; late fusion from scores (late soft fusion); and late fusion from decisions (late hard fusion). A theoretical analysis of the conditions for applying early or late fusion was included. It was demonstrated that if the available data samples are sufficient, i.e., if the a posteriori probability or score is known precisely, late fusion will not be able to outperform early fusion. Early fusion can lead to the curse of dimensionality, which can be alleviated by using dimension reduction methods such as principal component analysis. On the other hand, late fusion is the practical option in real scenarios, and it was demonstrated that late hard fusion will not be able to outperform late soft fusion. Nevertheless, determining the best option in a specific application context requires a strong experimental component. The specific comparison of weaknesses and strengthens of the state-of-the-art methods shows that, generally, the computational complexity is usually higher for early fusion, as it involves integrating information in the early stages of the data flow. Optimization may be easier to achieve in late fusion, as it focuses on scores and decisions, and accuracy, which may be more robust for specific contexts despite the loss of information. In summary, the choice of method to apply requires a thorough understanding of the context of the problem.

As a general conclusion, it is evident that fusion is a current topic in machine learning with significant room for continued research. Specifically, it is possible to suggest some lines of work. Firstly, a good part of the experimental work and most of the theoretical contributions only consider the two-class problem, the interest of an extension to an arbitrary number of classes being obvious. Along these lines, it should be mentioned that there is a great need for theoretical analyses that allow a better understanding of the very high number of experimental works existing in this area. For example, it would be most convenient to have an analysis of the different kinds of fusion when a posteriori probabilities are unknown and have to be estimated, as well as including in the theoretical analysis other factors such as computational complexity for real-time implementation. Another possible line of research could be called "fusion of fusions", that is, we could combine different levels of fusion, e.g., the output from an early feature fuser could be fused with the output of a soft late fuser. This could lead to the definition of complex fusion schemes. In a sense this is related to fusion in the context of deep learning, where the system itself can learn the best combination at different levels of the elements to be fused.

**Author Contributions:** The three researchers' contributions to this article are made equally in the following points: comparison between early fusion and late fusion, review of fusion methods at different levels, and literature search of works related to data fusion. All authors have read and agreed to the published version of the manuscript.

**Funding:** This research was funded by the Ministry of Science and Innovation of the Government of Spain MCIN/AEI/10.13039/501100011033 under Grants PRE2018-085092 and TEC2017-84743-P.

**Data Availability Statement:** Data are contained within the article.

**Conflicts of Interest:** The authors declare no conflicts of interest.

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
