# Peer review of "A Comparative Study on Recent Automatic Data Fusion Methodsâ€"

_computers, doi:10.3390/computers13010013_

Round 1
Reviewer 1 Report
Comments and Suggestions for Authors
The paper presents a comparative study of recent automatic data fusion methods, focusing on early and late fusion approaches. The authors provide a theoretical analysis of the conditions for applying early or late fusion and propose a comparative analysis with different fusion schemes, including the weaknesses and strengths of the state-of-the-art methods studied from the perspective of sensors, features, scores, and decisions. The study shows that if the available data samples are sufficient, late fusion will not be able to outperform early fusion. However, late fusion is the practical option in real-world scenarios, and late soft fusion is generally more robust than late hard fusion. The paper concludes that choosing which method to use requires a thorough understanding of the context of the problem.
The authors offer didactic toy examples of decision problems to illustrate the strengths and weaknesses of different fusion methods.
Figure 1 is puzzling. The number of papers published per year adds up to millions (e.g., 5.14 million in 2022). I don't understand what the authors are plotting in this figure, which is actually not very informative.
I am sure that the writing would be greatly improved if the senior authors would proofread the paper. In the corresponding box, I mention some points regarding the writing that the authors need to consider in their revision.
Comments on the Quality of English LanguageHere I mention some points regarding the writing that the authors need to consider in their revision.
Author Response
Reviewer 1 comments
Comment #1. “Figure 1 is puzzling. The number of papers published per year adds up to millions (e.g., 5.14 million in 2022). I don't understand what the authors are plotting in this figure, which is actually not very informative.
I am sure that the writing would be greatly improved if the senior authors would proofread the paper. In the corresponding box, I mention some points regarding the writing that the authors need to consider in their revision.”
Our response: Thank you for the comments and concerns. We agree that Figure 1 was puzzling and not very informative, thus we have removed it. In its place, we have included a new section “1.1 Data Fusion Concepts” where several basis concepts on data fusion are explained and two new figures are included. Sorry, we could not find the points regarding the writing, however, we have proofread the paper, finding and correcting some writing errata.
Reviewer 2 Report
Comments and Suggestions for Authors
A Comparative Study on Recent Automatic Data Fusion Methods focuses on the field of automatic data fusion in machine learning, aiming to enhance classification performance through various fusion methods.
The paper presents a comprehensive analysis of recent data fusion techniques, discussing early and late fusion strategies, including soft and hard fusion, and provides a theoretical analysis of these methods. The study includes a comparative analysis of different fusion schemes, assessing their strengths and weaknesses.
Comments and Suggestions to Authors:
- Clarity and Structure: The paper is well-structured, with clear distinctions between different types of fusion methods. However, it could benefit from a more detailed introduction to the field for readers unfamiliar with data fusion in machine learning.
- Theoretical Analysis: The theoretical analysis provides valuable insights, but it may be enhanced by including more real-world examples or case studies to illustrate the practical applications of these fusion methods.
- Comparative Analysis: The comparative analysis is thorough but could be more impactful if it included a discussion on the potential future developments or emerging trends in data fusion.
Author Response
Reviewer 2 comments
Comment #1. “The paper is well-structured, with clear distinctions between different types of fusion methods. However, it could benefit from a more detailed introduction to the field for readers unfamiliar with data fusion in machine learning.”
Comment #2. “The theoretical analysis provides valuable insights, but it may be enhanced by including more real-world examples or case studies to illustrate the practical applications of these fusion methods.”
Our response: Thank you for the comments and concerns. Regarding to comments 1 and 2, we have included the new section “1.1 Data Fusion Concepts” where several basis concepts on data fusion are explained considering readers unfamiliar with those topics. Besides, the explanation on several real-world examples is also extended in that section.
Comment #3. “The comparative analysis is thorough but could be more impactful if it included a discussion on the potential future developments or emerging trends in data fusion.”
Our response: Thank you for the comments and concerns. We have added the following discussion in the Conclusions section.
“As a general conclusion, it is evident that fusion is a current topic in machine learning with significant room for continued research. Specifically, it is possible to suggest some lines of work. Firstly, a good part of the experimental work and most of the theoretical contributions only consider the two-class problem, the interest of an extension to an arbitrary number of classes being obvious. Along these lines, it should be mentioned that there is a great need for theoretical analyses that allow a better understanding of the very high number of experimental works existing in this area. For example, it would be most convenient an analysis of the different kinds of fusion when posteriori probabilities are unknown and have to be estimated, as well as including in theoretical analysis other factors such as computational complexity for real-time implementation. Another possible line of research could be called “fusion of fusions”, that is, we could combine different levels of fusion, e.g. the output from and early feature fuser, could be fused with the output of a soft late fuser. This could lead to the definition of complex fusion schemes. In a sense this is related to fusion in the context of deep learning, where the system itself can learn the best combination at different levels of the elements to be fused.”